# RMFLOW: REFINED MEAN FLOW BY A NOISE-INJECTION STEP FOR MULTIMODAL GENERATION

**Yuhao Huang**[1], **Shih-Hsin Wang**[1], **Andrea L. Bertozzi**[2], **Bao Wang**[4] *

[1]Department of Mathematics and Scientific Computing and Imaging (SCI) Institute
University of Utah, Salt Lake City, UT, 84102, USA
[2]Department of Mathematics, UCLA, Los Angeles, CA, 90095, USA

## ABSTRACT

Mean flow (MeanFlow) enables efficient, high-fidelity image generation, yet its single-function evaluation (1-NFE) generation often cannot yield compelling results. We address this issue by introducing RMFlow, an efficient multimodal generative model that integrates a coarse 1-NFE MeanFlow transport with a subsequent tailored noise-injection refinement step. RMFlow approximates the average velocity of the flow path using a neural network trained with a new loss function that balances minimizing the Wasserstein distance between probability paths and maximizing sample likelihood. RMFlow achieves near state-of-the-art results on text-to-image, context-to-molecule, and time-series generation using only 1-NFE, at a computational cost comparable to the baseline MeanFlows.

## 1 INTRODUCTION

Flow matching (FM), closely related to diffusion models (DMs) (Sohl-Dickstein et al., 2015; Ho et al., 2020; Song et al., 2020), has emerged as a flexible framework for generative modeling, offering a principled way to learn transport between two distributions (cf. Lipman et al. (2023); Liu et al. (2023a); Albergo & Vanden-Eijnden (2023)). By approximating the instantaneous velocity field of this transport with a neural network, FM enables high-fidelity multimodal generation by solving the ordinary differential equation (ODE) with the neural network-approximated vector field as its forcing term (Esser et al., 2024; Ma et al., 2024; Polyak et al., 2024; Jing et al., 2024; Campbell et al., 2024). Nevertheless, this high-fidelity generation requires multiple expensive neural network evaluations, counted by the number of function evaluations (NFEs) (Chen et al., 2018).

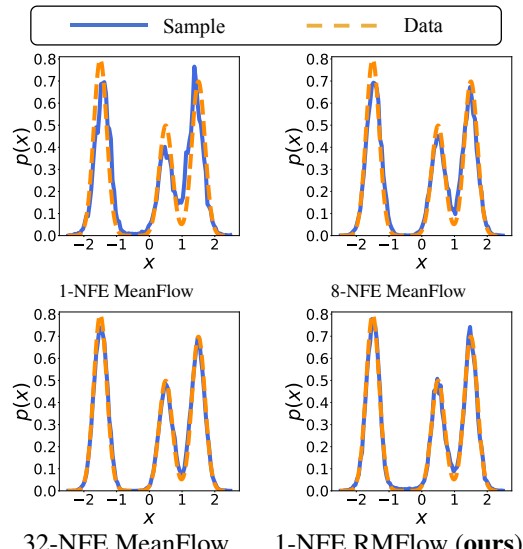

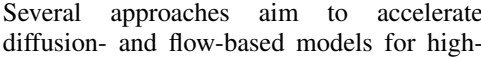

Figure 1: Contrasting MeanFlow with RMFlow for mixture Gaussian sampling; see Section 5.1 for experimental details and more results.

Several approaches aim to accelerate diffusion- and flow-based models for high-fidelity generation with only a few NFEs. Among these, consistency models (CMs) Song et al. (2023); Geng et al. (2024); Song & Dhariwal (2023); Lu & Song (2025) achieve remarkable performance and efficiency. Distillation is a noticeable idea; for instance, local FM (Xu et al., 2024a) breaks the flow into local sub-flows, enabling smaller models and easier distillation.

Recently, flow maps (Boffi et al., 2024; 2025) and mean flows (MeanFlows) (Geng et al. (2023); cf. Section 2) have been proposed to enable aggressive 1-NFE generation, and a prominent advan-

---

*Correspond to `wangbaonj@gmail.com`

tage of flow maps and MeanFlows is that they require no pre-training, distillation, or curriculum learning. Empirically, MeanFlows achieve high-quality image generation with fewer transport steps than FM models. However, preserving this quality typically requires multiple evaluations of the mean velocity field, as collapsing the process to 1-NFE often causes significant performance degradation. We showcase this issue by sampling a mixture Gaussian distribution using MeanFlow; see Section 5.1 for experimental details. Figure 1 shows the significant gap between exact (data) and sampled distributions when using 1-NFE MeanFlow, and this gap reduces as NFE increases.

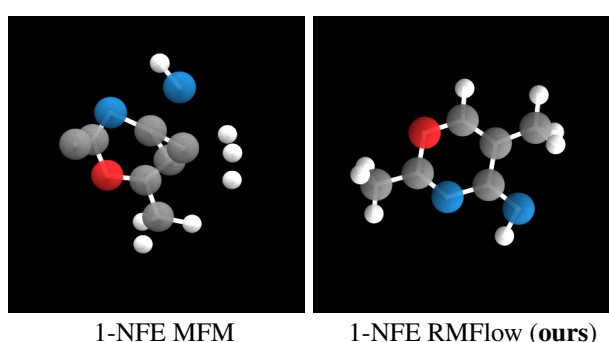

1-NFE MFM      1-NFE RMFlow (**ours**)

Figure 2: Contrasting MeanFlow with RMFlow, under the same context, for QM9 molecule generation.

We further showcase the significant generation error of 1-NFE MeanFlow for the benchmark QM9 molecule generation (Ramakrishnan et al., 2014); see Section 5.2 for experimental details and additional results. Figure 2 illustrates that 1-NFE MeanFlow produces an invalid structure, where the molecule is fragmented into multiple disconnected pieces. Indeed, in our experiments, we consistently observed that 1-NFE MeanFlow frequently generates invalid structures. Additional quantitative results in Section 5.2 further confirm the significant errors associated

with 1-NFE MeanFlow generation.

The above numerical results motivate us to study the following problem:

*Can we improve the performance of 1-NFE MeanFlows for multimodal generation?*

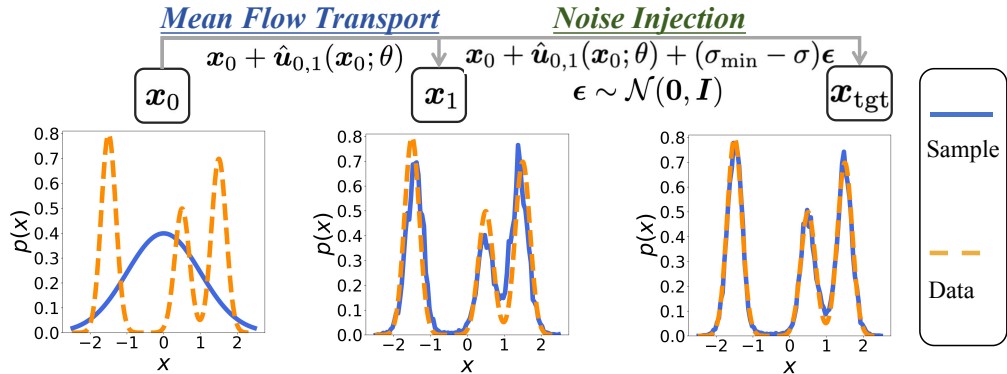

Figure 3: Schematic of our proposed RMFlow: it first applies 1-NFE MeanFlow transport, then refines the result by a subsequent noise-injection step; see Section 3. The average velocity $\hat{\boldsymbol{u}}_{0,1}(\boldsymbol{x}_0; \theta)$ of RMFlow is trained by incorporating the maximum likelihood objective into the MeanFlow framework, as in equation 12.

## 1.1 OUR CONTRIBUTIONS

We propose RMFlow—an improved 1-NFE MeanFlow model for multimodal generation. RMFlow leverages 1-NFE MeanFlow for coarse transport, accompanied by a subsequent tailored noise-injection step to refine the generation; Fig. 3 depicts the idea of RMFlow. The results in Figs 1 and 2 demonstrate that RMFlow achieves a substantial improvement in generation quality over 1-NFE MeanFlow. In particular, it effectively mitigates invalid structures, producing coherent and valid molecular graphs. Our key contributions are:

- We propose RMFlow to enable 1-NFE high-fidelity multimodal generation by integrating the guidance encoding with a tailored noise-injection refinement strategy; see Section 3.

- We design a theoretically principled training objective for RMFlow that balances minimizing the Wasserstein distance between probability paths and maximizing the likelihood of the learned target distribution; see Section 4.

- We show the compelling, often (near) state-of-the-art, results of RMFlow for benchmark text-to-image, text-to-structure, and time series generation (see Section 5).

## 1.2 ADDITIONAL RELATED WORKS

To our knowledge, this work is the first to improve MeanFlows by introducing a noise-injection refinement for 1-NFE generation. This differs from existing couplings of flow and diffusion models, such as Diff2Flow (Schusterbauer et al., 2025), which transfers knowledge from pretrained diffusion models to flow matching models, and generator matching (Patel et al., 2024), which connects diffusion and flow matching under Markov generative processes.

Another line of work studies error control in FM. Prior analyses of probability flow ODEs and FM (Song et al., 2021; Lu et al., 2022; Lai et al., 2023; Albergo et al., 2023) show that FM alone cannot guarantee likelihood maximization or KL divergence minimization between target and learned distributions.

## 1.3 ORGANIZATION

We organize the rest of this paper as follows: We provide necessary background materials in Section 2. We present our proposed 1-NFE RMFlow in Section 3. We present our training loss function for RMFlow in Section 4. Our numerical results for RMFlow in Section 5. Technical proofs and additional experimental details and results are provided in the appendix.

## 2 BACKGROUND

In this section, we provide a brief review of flow-based generative models, especially MeanFlows. For a detailed exploration of FM, we refer the reader to (Lipman et al., 2023; Liu et al., 2023a; Fukumizu et al., 2024). For a given data $\boldsymbol{x}_1 = \boldsymbol{x}_{\text{data}} \sim p$ and a prior sample $\boldsymbol{x}_0 \sim q$ (e.g., standard Gaussian $\mathcal{N}(\boldsymbol{0}, \boldsymbol{I})$), a (conditional) flow path—connecting the two samples—can be constructed as $\boldsymbol{x}_t = a_t \boldsymbol{x}_1 + b_t \boldsymbol{x}_0$ with $a_t$ and $b_t$ being predefined schedules. A common choice is $a_t = 1 - t$ and $b_t = t$, which corresponds to rectified flow (Liu et al., 2023a). This interpolation can be equivalently expressed as the solution to the ODE $\dot{\boldsymbol{x}}_t = \boldsymbol{u}_t(\boldsymbol{x}_t|\boldsymbol{z})$, where $\boldsymbol{z} = (\boldsymbol{x}_0, \boldsymbol{x}_1)$ denotes the coupling of start and end points, and $\boldsymbol{u}_t(\boldsymbol{x}_t|\boldsymbol{z}) = \dot{a}_t \boldsymbol{x}_1 + \dot{b}_t \boldsymbol{x}_0$ is the conditional vector field. FM learns an unconditional vector field $\boldsymbol{u}_t(\boldsymbol{x}) := \mathbb{E}_{\boldsymbol{z}}[\boldsymbol{u}_t(\boldsymbol{x}|\boldsymbol{z})|\boldsymbol{x}_t = \boldsymbol{x}]$, which does not require knowledge of the pair $\boldsymbol{z} = (\boldsymbol{x}_0, \boldsymbol{x}_1)$. This is achieved by training a neural network $\hat{\boldsymbol{u}}_t(\boldsymbol{x}; \theta)$ to minimize the objective:

$$\mathcal{L}_{\text{CFM}}(\theta) := \mathbb{E}_{t,\boldsymbol{z}}\left[\|\hat{\boldsymbol{u}}_t(\boldsymbol{x}_t; \theta) - \boldsymbol{u}_t(\boldsymbol{x}_t|\boldsymbol{z})\|^2\right]. \tag{1}$$

After training, we generate data by integrating $\frac{d\boldsymbol{x}_t}{dt} = \hat{\boldsymbol{u}}_t(\boldsymbol{x}_t; \theta)$ from $t = 0$ to $1$, with $\boldsymbol{x}_0 \sim q$.

Although FM is conceptually simple, sample generation requires multiple evaluations of $\hat{\boldsymbol{u}}_t(\boldsymbol{x}_t; \theta)$, which can be computationally intensive. To address this inefficiency issue, MeanFlow learns an averaged velocity field based on the instantaneous velocity field $\boldsymbol{u}_t(\boldsymbol{x}_t)$, defined as:

$$\boldsymbol{u}_{t,r}(\boldsymbol{x}_t) := \frac{\boldsymbol{x}_r - \boldsymbol{x}_t}{r - t} = \frac{1}{r - t}\int_t^r \boldsymbol{u}_s(\boldsymbol{x}_s)ds. \tag{2}$$

This allows data generation by transporting $\boldsymbol{x}_t$ to $\boldsymbol{x}_r$ using the approximate average velocity field $\hat{\boldsymbol{u}}_{t,r}$:

$$\boldsymbol{x}_r = \boldsymbol{x}_t + (r - t)\hat{\boldsymbol{u}}_{t,r}(\boldsymbol{x}_t; \theta). \tag{3}$$

In particular, 1-NFE generation corresponds to $\boldsymbol{x}_1 = \boldsymbol{x}_0 + \hat{\boldsymbol{u}}_{0,1}(\boldsymbol{x}_0; \theta)$. For Multi-NFE generation, $\hat{\boldsymbol{u}}_{t,r}(\boldsymbol{x}; \theta)$ is evaluated sequentially on a chosen grid $0 = \tau_0 < \cdots < \tau_n = 1$ and is applied between consecutive grid points to transport samples. This approach achieves high-fidelity generation with significantly fewer NFEs compared to FM models that rely on the instantaneous velocity field $\boldsymbol{u}_t(\boldsymbol{x})$.

In MeanFlows, the mean velocity field $\boldsymbol{u}_{t,r}(\boldsymbol{x})$ is approximated by a neural network $\hat{\boldsymbol{u}}_{t,r}(\boldsymbol{x};\theta)$, with the weights $\theta$ being calibrated by minimizing the following conditional mean flow matching $\mathcal{L}_{\text{CMFM}}$ loss function:

$$\mathcal{L}_{\text{CMFM}}(\theta) := \mathbb{E}_{t,r,\boldsymbol{z}}\left[\|\hat{\boldsymbol{u}}_{t,r}(\boldsymbol{x};\theta) - \text{sg}\left(\boldsymbol{u}_{t,r}^{\text{tgt}}(\boldsymbol{x};\theta)\right)\|^2\right], \tag{4}$$

where $0 \leq t \leq r \leq 1$ are uniform samples from the interval $[0, 1]$ and $\boldsymbol{u}_{t,r}^{\text{tgt}}$ is the target defined as:

$$\boldsymbol{u}_{t,r}^{\text{tgt}}(\boldsymbol{x};\theta) := \boldsymbol{u}_t(\boldsymbol{x}|\boldsymbol{z}) + (r - t)\left[\nabla\hat{\boldsymbol{u}}_{t,r}(\boldsymbol{x};\theta) \cdot \boldsymbol{u}_t(\boldsymbol{x}|\boldsymbol{z}) + \partial_t\hat{\boldsymbol{u}}_{t,r}(\boldsymbol{x};\theta)\right],$$

with sg denoting a stop-gradient operation. This stop-gradient approach prevents higher-order optimization while ensuring that zero loss guarantees dynamical consistency. The target velocity $\boldsymbol{u}_{t,r}^{\text{tgt}}$ is efficiently computed using Jacobian-vector products (jvp) in autodiff libraries such as PyTorch (Paszke et al., 2019) or JAX (Bradbury et al., 2018).

# 3 THE DESIGN OF RMFLOW

In this section, we describe the design of 1-NFE RMFlow for high-fidelity generation, with or without multimodal guidance.

## 3.1 MOTIVATION

In practice, the true data distribution $\boldsymbol{x}_{\text{data}}$ is unavailable due to its complexity and the limited nature of observed data. Following the standard practice in the field, as established in works such as (Ho et al., 2020; Song et al., 2021; Lu et al., 2022; Lipman et al., 2023), we approximate it with a noisy, smoothed version $\boldsymbol{x}_{\text{tgt}} = \boldsymbol{x}_{\text{data}} + \sigma_{\min}\boldsymbol{\epsilon} \sim p_{\text{tgt}}$, where $\boldsymbol{\epsilon} \sim \mathcal{N}(\mathbf{0}, \boldsymbol{I})$ and $\sigma_{\min}$ is small (e.g., $10^{-3}$). This approach ensures stability and robust learning of the data distribution.

MeanFlows learn a neural network, by minimizing $\mathcal{L}_{\text{CMFM}}$ in equation 4, to transport a prior sample $\boldsymbol{x}_0$ directly to the noisy target, i.e., $\boldsymbol{x}_1 = \boldsymbol{x}_{\text{tgt}}$. Boffi et al. (2024; 2025) showed that this approach reduces the Wasserstein distance between the target distribution $p_{\text{tgt}}$ and the learned distribution $p_\theta$:

**Theorem 3.1.** *[Boffi et al. (2025)] There exists a constant $M > 0$ such that:*

$$M \cdot \mathcal{L}_{\text{CMFM}}(\theta) \geq W_2^2(p_{\text{tgt}}, p_\theta) := \inf_{\gamma \in \Pi(p_{\text{tgt}}, p_\theta)} \mathbb{E}_{(x,y)\sim\gamma}\left[\|x - y\|^2\right], \tag{5}$$

*where $M$ is a constant, $W_2^2(p_{\text{tgt}}, p_\theta)$ denotes the Wasserstein distance between $p_{\text{tgt}}$ and $p_\theta$, and $\Pi(p_{\text{tgt}}, p_\theta)$ is the set of all joint distributions with marginals $p_{\text{tgt}}$ and $p_\theta$.*

While controlling the Wasserstein distance provides a meaningful measure of distributional alignment, empirical evidence indicates that FM enforces additional constraints, such as KL divergence (Lu et al., 2022), often achieves superior generative performance over the FM baseline. With this in mind, we aim to enhance the fidelity of 1-NFE MeanFlows, pushing beyond current limitations in a manner analogous to improvements seen in FMs.

## 3.2 NOISE INJECTION REFINEMENT

We decompose the generation process into two stages. In the first stage, a 1-NFE MeanFlow transports the prior $\boldsymbol{x}_0$ to an intermediate noisy state

$$\boldsymbol{x}_1 = \boldsymbol{x}_{\text{data}} + \sigma\epsilon_1, \quad \text{with } \epsilon_1 \sim \mathcal{N}(\mathbf{0}, \boldsymbol{I}) \text{ and } \sigma < \sigma_{\min}. \tag{6}$$

In the second stage, a *single noise injection step* is applied:

$$\boldsymbol{x}_{\text{tgt}} = \boldsymbol{x}_1 + \sqrt{\sigma_{\min}^2 - \sigma^2} \cdot \epsilon_2, \quad \epsilon_2 \sim \mathcal{N}(\mathbf{0}, \boldsymbol{I}), \tag{7}$$

to generate the final sample. This additional noise injection aligns with the designs of VAEs (Kingma & Welling, 2013), allowing principled likelihood maximization via a loss term derived from the evidence lower bound (ELBO) (Wainwright et al., 2008) to optimize the MeanFlow parameters. We will prove in Theorem 4.1 that this formulation enables control over the KL divergence between the target distribution $p_{\text{tgt}}$ and the learned distribution $p_\theta$.

In summary, **our data generation process** is defined as

$$\hat{\boldsymbol{x}}_{\text{tgt}} = \boldsymbol{x}_0 + \hat{\boldsymbol{u}}_{0,1}(\boldsymbol{x}_0; \theta) + \sqrt{\sigma_{\min}^2 - \sigma^2} \cdot \epsilon_2, \quad \epsilon_2 \sim \mathcal{N}(\boldsymbol{0}, \boldsymbol{I}), \tag{8}$$

where $\hat{\boldsymbol{u}}_{0,1}(\boldsymbol{x}_0; \theta)$ denotes the learned average velocity field. Although RMFlow is conceptually a two-stage framework, equation 8 demonstrates that generation is performed in a single step: the learned flow is evaluated once (1-NFE), and a noise term is added in parallel to produce the output.

### 3.3 MULTIMODALITY

To support cross-modality generation, we incorporate an encoder $\phi_\omega(\boldsymbol{c})$ that embeds conditioning signals (e.g., text prompts). The prior samples for both guided (potentially multimodal) and unguided generation are defined as

$$\boldsymbol{x}_0 = \begin{cases} \phi_\omega(\boldsymbol{c}) + \sigma_c \epsilon, & \text{for guided generation,} \\ \epsilon, & \text{for unguided generation,} \end{cases}$$

where $\epsilon \sim \mathcal{N}(\boldsymbol{0}, \boldsymbol{I})$. This design allows the flow to incorporate multimodal guidance if available, while defaulting to unconditional generation otherwise. Here, $\phi_\omega(\cdot)$ is an encoder chosen following common practice (see Section 5), and $\sigma_c \ll 1$ (e.g., $10^{-3}$) is pre-chosen to control perturbations.

Specifically, for a given data pair $(\boldsymbol{x}_{\text{data}}, \boldsymbol{c})$, we train the MeanFlow to transport the prior sample $\boldsymbol{x}_0 = \phi_\omega(\boldsymbol{c}) + \sigma_c \epsilon$ to the intermediate target $\boldsymbol{x}_1 = \boldsymbol{x}_{\text{data}} + \sigma \epsilon_1$, where $\epsilon_1 \sim \mathcal{N}(\boldsymbol{0}, \boldsymbol{I})$. *The encoder and MeanFlow are optimized jointly*, and we will discuss the training objective in Section 4.

**Remark 1.** *Our proposed RMFlow differs from MeanFlow in two aspects: (1) We apply a tailored encoder to the guidance, (2) we add a noise injection step to refine the generation result.*

## 4 THE TRAINING OF RMFLOW

In this section, we present the training procedure for RMFlow. We first establish the theoretical foundation of noise-injection refinement, showing that it enables likelihood maximization of the learned distribution with respect to the target distribution. Building on this, we introduce a joint training objective that combines $\mathcal{L}_{\text{CMFM}}$ (for Wasserstein control) with likelihood maximization and optional guidance regularization, ensuring both fidelity and flexibility in guided generation. Finally, we adopt parameter-efficient fine-tuning (PEFT; cf. (Hu et al., 2022; Dettmers et al., 2023)) to implement RMFlow for large-scale tasks.

### 4.1 LIKELIHOOD MAXIMIZATION

In this section, we show that the noise-injection step in equation 8 enables likelihood maximization during RMFlow training. Specifically, for a given prior sample $\boldsymbol{x}_0$, the intermediate sample generated by the MeanFlow is

$$\boldsymbol{x}_1 = \boldsymbol{x}_0 + \hat{\boldsymbol{u}}_{0,1}(\boldsymbol{x}_0; \theta).$$

By equation 8, the conditional distribution of the final generated sample given the prior is

$$\hat{\boldsymbol{x}}_{\text{tgt}} \mid \boldsymbol{x}_0 \sim \mathcal{N}\Big(\boldsymbol{x}_0 + \hat{\boldsymbol{u}}_{0,1}(\boldsymbol{x}_0; \theta), (\sigma_{\min}^2 - \sigma^2)\boldsymbol{I}\Big).$$

This specifies a parametric conditional distribution. Given an observed target $\boldsymbol{x}_{\text{tgt}}$, the corresponding conditional log-likelihood is

$$\log p_\theta(\boldsymbol{x}_{\text{tgt}} \mid \boldsymbol{x}_0) = -\frac{1}{2((\sigma_{\min}^2 - \sigma^2))} \big\| \boldsymbol{x}_{\text{tgt}} - \big(\boldsymbol{x}_0 + \hat{\boldsymbol{u}}_{0,1}(\boldsymbol{x}_0; \theta)\big) \big\|^2 + C, \tag{9}$$

where $C = -\frac{d}{2} \log(2\pi(\sigma_{\min}^2 - \sigma^2))$ and $d$ is the dimensionality of the data. Notice that equation 6 and equation 7 indicate the observed target $\boldsymbol{x}_{\text{tgt}} = \boldsymbol{x}_{\text{data}} + \sigma_{\min} \epsilon$, where $\epsilon \sim \mathcal{N}(\boldsymbol{0}, \boldsymbol{I})$.

Therefore, we define the following loss term to maximize the likelihood:

$$\mathcal{L}_{\text{NLL}} := \mathbb{E}_{\boldsymbol{x}_0, \boldsymbol{x}_{\text{data}}, \epsilon} \Big[ \big\| (\boldsymbol{x}_{\text{data}} + \sigma_{\min} \epsilon) - (\boldsymbol{x}_0 + \hat{\boldsymbol{u}}_{0,1}(\boldsymbol{x}_0; \theta)) \big\|^2 \Big]. \tag{10}$$

The following theorem formalizes the theoretical guarantee of the noise-injection refinement. In particular, it demonstrates that minimizing the loss $\mathcal{L}_{\mathrm{NLL}}$ maximizes the expected log-likelihood, thereby reducing the KL divergence between the target and learned distributions.

**Theorem 4.1.** *The negative log-likelihood loss $\mathcal{L}_{\mathrm{NLL}}$ provides a lower bound on the expected log-likelihood of the target distribution:*

$$-A \cdot \mathcal{L}_{\mathrm{NLL}} + C \leq \mathbb{E}_{\boldsymbol{x}_{\mathrm{tgt}}}[\log p_\theta(\boldsymbol{x}_{\mathrm{tgt}})] = -H(p_{\mathrm{tgt}}) - D_{\mathrm{KL}}(p_{\mathrm{tgt}} \,\|\, p_\theta), \tag{11}$$

*where $H(p_{\mathrm{tgt}}) := -\mathbb{E}_{\boldsymbol{x}_{\mathrm{tgt}}}[\log p_{\mathrm{tgt}}]$ is the entropy of $p_{\mathrm{tgt}}$, $D_{\mathrm{KL}}(p_{\mathrm{tgt}}\|p_\theta) := \mathbb{E}_{\boldsymbol{x}_{\mathrm{tgt}}}[\log \frac{p_{\mathrm{tgt}}}{p_\theta}]$ denotes the KL divergence between the target and the learned distributions, and $A, B > 0$ are constants.*

### 4.1.1 JOINT TRAINING OBJECTIVE

RMFlow is trained by jointly optimizing the original MeanFlow loss (Wasserstein control) and likelihood maximization, resulting in the following objective function:

$$\mathcal{L}_{\mathrm{RMFlow}}(\theta, \omega) = \underbrace{\mathcal{L}_{\mathrm{CMFM}}}_{\mathrm{I}} + \underbrace{\lambda_1 \mathcal{L}_{\mathrm{NLL}}}_{\mathrm{II}} + \underbrace{\lambda_2 \mathbb{E}_{(\boldsymbol{x}_{\mathrm{data}}, \boldsymbol{c})}[\|\phi_\omega(\boldsymbol{c})\|^2]}_{\mathrm{III}}, \tag{12}$$

where $\lambda_1, \lambda_2 \geq 0$ are two hyperparameters. We remark that Term I controls the gap between the probability flows of the exact and approximated mean velocities in intermediate states, Term II for likelihood maximization, and Term III is designed for guided generation and is set to 0 for unguided generation. Here, the expectation in III is taken over all data-guidance pairs $(\boldsymbol{x}_{\mathrm{data}}, \boldsymbol{c})$.

**Remark 2.** *Term III in equation 12 can be considered as a regularization on the prior distribution, and a similar term is used in training VAE (Kingma & Welling, 2013). Empirically, we observe that term III can be very large, resulting in substantial performance degradation.*

## 4.2 MEMORY-EFFICIENT FINE-TUNING

For relatively small-scale tasks, we train our RMFlow by directly minimizing $\mathcal{L}_{\mathrm{RMFlow}}$. Compared to $\mathcal{L}_{\mathrm{CMFM}}$, our new objective $\mathcal{L}_{\mathrm{RMFlow}}$ introduces additional gradient pathways, increasing memory footprint. To balance efficiency and performance for large-scale tasks, we first train the MeanFlow model by minimizing $\mathcal{L}_{\mathrm{CMFM}}$, and then fine-tune it using PEFT (Hu et al., 2022; Dettmers et al., 2023), with $\mathcal{L}_{\mathrm{RMFlow}}$ as a supervised objective in our large-scale experiments on text-to-image and molecule generation tasks. During fine-tuning, we further strengthen training by integrating 1-NFE sampling with a policy-gradient objective that incorporates physical feedback on sample quality for molecule generation tasks, as described in Zhou et al. (2025).

## 5 NUMERICAL EXPERIMENTS

In this section, we validate the efficacy and efficiency of RMFlow for both guided and unguided sample generation. We consider two synthetic tasks: sampling a 1D mixture Gaussian distribution and a 2D checkerboard density (Section 5.1). We also consider several benchmark tasks, including context-to-molecular structure generation (Section 5.2), sampling trajectories of dynamical systems (time series; Section 5.3), and text-to-image generation (Section 5.4).

**Software and Equipment.** We implement synthetic tasks, context-to-molecule generation, and text-to-image generation using `PyTorch`. We implement the time series generation task using `JAX`. Additionally, we use `Torch DDP` and `torch.compile` to optimize the model execution for context-to-molecule and text-to-image generation. All the experiments are carried out on multiple NVIDIA RTX 3090/4090 GPUs.

**Training Setups.** See Appendix B for the details of training setups.

**Evaluation Metrics:** For synthetic tasks and time-series generation, we evaluate performance using the estimated KL divergence and total variation (TV) distance between the generated samples and the ground-truth. Both KL and TV are computed from densities obtained via histogram-based estimation of the sample and ground-truth distributions. For molecule generation, we predict bond types from pairwise interatomic distances and atom types, and then compute atom and molecule stability,

following Hoogeboom et al. (2022). For the image generation task, we assess sample quality using the Fréchet Inception Distance (FID) (Heusel et al., 2017).

We use NFE to measure generation efficiency following (Geng et al., 2025). Notice that the Gaussian noise injection step takes negligible time compared to the neural network function evaluation.

|    | 1-NFE MeanFlow | 8-NFE MeanFlow | 32-NFE MeanFlow | 1-NFE RMFlow (**ours**) |
|----|----------------|----------------|-----------------|-------------------------|
| TV | 1.4422         | 0.7977         | 0.6737          | 0.7567                  |
| KL | 0.8074         | 0.4074         | 0.1017          | 0.2332                  |

Table 1: Contrasting 1-NFE RMFlow with 1/8/32-NFE MeanFlow for mixture Gaussian sampling. 1-NFE RMFlow outperforms both 1- and 8-NFE MeanFlows, while slightly worse than 32-NFE MeanFlow.

|    | 1-NFE MeanFlow | 8-NFE MeanFlow | 32-NFE MeanFlow | 1-NFE RMFlow (**ours**) |
|----|----------------|----------------|-----------------|-------------------------|
| TV | 0.238          | 0.167          | 0.155           | 0.173                   |
| KL | 0.311          | 0.139          | 0.118           | 0.163                   |

Table 2: Contrasting 1-NFE RMFlow with 1/8/32-NFE MeanFlow for checkerboard density sampling. 1-NFE RMFlow significantly outperforms 1-NFE MeanFlow, closing the performance gap to multi-NFE MeanFlow.

## 5.1 SYNTHETIC TASKS

In this experiment, we train a simple ResNet-based model under both MeanFlow and RMFlow frameworks for $10^5$ iterations using a batch size of 256 to sample (1) 1D Gaussian mixture $p_{\text{data}} = 0.35\mathcal{N}(1.5, 0.04) + 0.25\mathcal{N}(0.5, 0.04) + 0.4\mathcal{N}(-1.5, 0.04)$, and (2) 2D checkerboard where the probability density resembles a checkerboard pattern. We consider 1/8/32-NFE MeanFlow and 1-NFE RMFlow for sample generation. Tables 1 and 2 show that 1-NFE RMFlow significantly outperforms 1-NFE MeanFlow, closing the performance gap to multi-NFE MeanFlow.

## 5.2 CONTEXT-TO-MOLECULE: QM9 GENERATION

We train MeanFlow and RMFlow for context-to-molecule generation on the QM9 dataset Ramakrishnan et al. (2014), a benchmark containing atomic coordinates and quantum-chemical properties for 130k small molecules with up to 9 heavy atoms (up to 29 atoms including hydrogens). Following Hoogeboom et al. (2022), we perform condition generation on seven molecular properties: (1) number of atoms, (2) HOMO, (3) LUMO, (4) $\alpha$ (isotropic polarizability), (5) gap, (6) $\mu$ (dipole moment), and (7) $C_v$ (heat capacity). These properties are concatenated into a context vector and mapped to the data space using $\phi_\omega(c)$, parameterized by a single EGNN block Garcia Satorras et al. (2021).

Our model backbone follows the EGNN architecture in (Garcia Satorras et al., 2021; Hoogeboom et al., 2022), augmented with a time-embedding module for the additional scalar time variable $r$. In addition, molecule stability is used as the reward within the RL framework, following the approach of Zhou et al. (2025), to provide feedback during training (see Section 4.2 and Appendix B.8). We adopt the train/val/test splits of Anderson et al. (2019), comprising $100k/18k/13k$ molecules, respectively. Table 3 shows that 1-NFE RMFlow attains state-of-the-art performance, whereas competing SOTA methods require $n$-NFE with $n \gg 1$. Figure 4 depicts a few randomly generated molecules and the corresponding contexts.

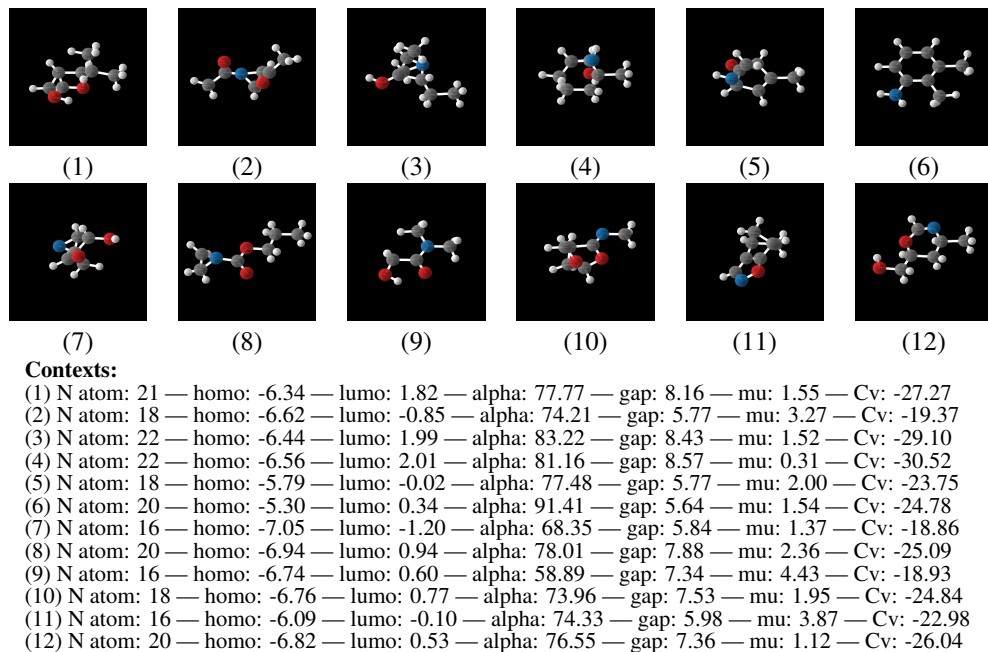

**Contexts:**
(1) N atom: 21 — homo: -6.34 — lumo: 1.82 — alpha: 77.77 — gap: 8.16 — mu: 1.55 — Cv: -27.27
(2) N atom: 18 — homo: -6.62 — lumo: -0.85 — alpha: 74.21 — gap: 5.77 — mu: 3.27 — Cv: -19.37
(3) N atom: 22 — homo: -6.44 — lumo: 1.99 — alpha: 83.22 — gap: 8.43 — mu: 1.52 — Cv: -29.10
(4) N atom: 22 — homo: -6.56 — lumo: 2.01 — alpha: 81.16 — gap: 8.57 — mu: 0.31 — Cv: -30.52
(5) N atom: 18 — homo: -5.79 — lumo: -0.02 — alpha: 77.48 — gap: 5.77 — mu: 2.00 — Cv: -23.75
(6) N atom: 20 — homo: -5.30 — lumo: 0.34 — alpha: 91.41 — gap: 5.64 — mu: 1.54 — Cv: -24.78
(7) N atom: 16 — homo: -7.05 — lumo: -1.20 — alpha: 68.35 — gap: 5.84 — mu: 1.37 — Cv: -18.86
(8) N atom: 20 — homo: -6.94 — lumo: 0.94 — alpha: 78.01 — gap: 7.88 — mu: 2.36 — Cv: -25.09
(9) N atom: 16 — homo: -6.74 — lumo: 0.60 — alpha: 58.89 — gap: 7.34 — mu: 4.43 — Cv: -18.93
(10) N atom: 18 — homo: -6.76 — lumo: 0.77 — alpha: 73.96 — gap: 7.53 — mu: 1.95 — Cv: -24.84
(11) N atom: 16 — homo: -6.09 — lumo: -0.10 — alpha: 74.33 — gap: 5.98 — mu: 3.87 — Cv: -22.98
(12) N atom: 20 — homo: -6.82 — lumo: 0.53 — alpha: 76.55 — gap: 7.36 — mu: 1.12 — Cv: -26.04

Figure 4: A few randomly selected RMFlow-generated molecules, together with the corresponding contexts.

| Metrics | No Tuncation Error (Discretization) | Atomic Stab. ($\uparrow$) | Mol Stab. ($\uparrow$) | NFE($\downarrow$) |
|---|---|---|---|---|
| ENF | ✗ | $85_{\pm 0.1}\%$ | $4.9_{\pm 0.2}\%$ | $\gg 1$ |
| E-DM Hoogeboom et al. (2022) | ✗ | $98.73_{\pm 0.1}\%$ | $82.11_{\pm 0.4}\%$ | $\gg 1$ |
| Bridge Wu et al. (2022) | ✗ | $98.7_{\pm 0.1}\%$ | $81.8_{\pm 0.2}\%$ | $\gg 1$ |
| Bridge + Force Wu et al. (2022) | ✗ | $98.8_{\pm 0.1}\%$ | $84.6_{\pm 0.3}\%$ | $\gg 1$ |
| GeoLDM Xu et al. (2023) | ✗ | $98.73\%$ | $89.40_{\pm 0.5}\%$ | $\gg 1$ |
| GeoBFN Song et al. (2024) | ✗ | $99.0\%$ | $93.9\%$ | $\gg 1$ |
| E-DM + RLPF Zhou et al. (2025) | ✗ | $99.1\%$ | $93.4\%$ | $\gg 1$ |
| MeanFlow w/o contexts | ✓ | $98.2_{\pm 0.07}\%$ | $79.3_{\pm 0.8}\%$ | 1 |
| MeanFlow w/ contexts | ✓ | $98.4_{\pm 0.05}\%$ | $84.3_{\pm 0.5}\%$ | 1 |
| RMFlow w/o contexts (**ours**) | ✓ | $98.8_{\pm 0.05}\%$ | $90.1_{\pm 0.5}\%$ | 1 |
| RMFlow w/ contexts (**ours**) | ✓ | $98.9_{\pm 0.05}\%$ | $93.2_{\pm 0.4}\%$ | 1 |
| RMFlow w/ contexts + RLPF (**ours**) | ✓ | $98.9_{\pm 0.05}\%$ | $93.5_{\pm 0.3}\%$ | 1 |
| Data | – | $99\%$ | $95.2\%$ | – |

Table 3: Contrasting the performance of different models for QM9 molecule generation. We run RMFlow with contexts by randomly selecting $10^4$ contexts in the test dataset of QM9 five times.

## 5.3 TIME SERIES: DYNAMICAL SYSTEM

Sampling trajectories in dynamical systems under event guidance is a key challenge for predicting and understanding complex phenomena such as climate and extreme events (Perkins & Alexander, 2013; Mosavi et al., 2018). Recent works (Finzi et al., 2023) and (Huang et al., 2025) have introduced diffusion and FM models specifically designed for event-guided sampling.

In this experiment, we perform dynamical system trajectory forecasting with MeanFlow and RM-Flow, formulating it as a time series problem by discretizing the time variable $t$ into uniform intervals. Each trajectory (either from a dataset or sampled) is a discrete time series of vectors concatenated into $\boldsymbol{x}_{\text{data}} = [\boldsymbol{x}(\tau_m)]_{m=1}^{M} \in \mathbb{R}^{Md}$, where $M$ is the total number of time steps, $d$ is the dimension of the system, and $\boldsymbol{x}(\tau_m) \in \mathbb{R}^d$ denotes the discretized trajectory at time $\tau_m$. Our goal is to generate $\boldsymbol{x}_{\text{data}} = [\boldsymbol{x}(\tau_m)]_{m=1}^{M} \in \mathbb{R}^{Md}$ with 1-NFE using MeanFlow and RMFlow.

We train our models on the Lorenz and FitzHugh–Nagumo dynamical systems (see (Huang et al., 2025, Appendix B.1) for a brief review of these two models); using a U-Net backbone. For event guidance, where events are defined by a constraint function $E = \{\boldsymbol{x}_{\text{data}} \,|\, C(\boldsymbol{x}_{\text{data}}) > 0\}$, we adopt a simple but effective design: the event-guidance vector and the first three states $\boldsymbol{x}(\tau_1), \boldsymbol{x}(\tau_2), \boldsymbol{x}(\tau_3)$ are embedded through an MLP $\phi_\omega$ into the target data space $\mathbb{R}^{Md}$. This avoids reliance on Tweedie's

formula as used in (Finzi et al., 2023; Huang et al., 2025). Tables 4 and 5 show that RMFlow yields significantly better 1-NFE generation than MeanFlow, while achieving accuracy comparable to multi-NFE methods.

| | Lorenz | | FitzHugh-Nagumo | | |
|---|---|---|---|---|---|
| Model | w/o $E$ ($\downarrow$) | w/ $E$ ($\downarrow$) | w/o $E$ ($\downarrow$) | w/ $E$ ($\downarrow$) | NFE ($\downarrow$) |
| Diffusion Huang et al. (2025) | 0.0314 | 0.1001 | 0.0277 | 0.1192 | 128 |
| FM Huang et al. (2025) | 0.0348 | 0.0972 | 0.0314 | 0.2164 | 128 |
| FDM Huang et al. (2025) | 0.0306 | 0.0914 | 0.0266 | 0.1168 | 128 |
| MeanFlow | 0.0469 | 0.1250 | 0.0398 | 0.2268 | 1 |
| MeanFlow | 0.0366 | 0.1011 | 0.0345 | 0.1988 | 8 |
| MeanFlow | 0.0351 | 0.0991 | 0.0302 | 0.1723 | 32 |
| RMFlow (**ours**) | 0.0332 | 0.0956 | 0.0289 | 0.1543 | 1 |

Table 4: TV distance between the generated (by different models) and test trajectory distributions, estimated from histogram-based density approximations, with/without conditioning on the event.

| | Lorenz | | FitzHugh-Nagumo | | |
|---|---|---|---|---|---|
| Model | w/o $E$ ($\downarrow$) | w/ $E$ ($\downarrow$) | w/o $E$ ($\downarrow$) | w/ $E$ ($\downarrow$) | NFE ($\downarrow$) |
| Diffusion Huang et al. (2025) | 0.0056 | 0.2774 | 0.0260 | 0.3011 | 128 |
| FM Huang et al. (2025) | 0.0081 | 0.2560 | 0.0280 | 0.3468 | 128 |
| FDM Huang et al. (2025) | 0.0049 | 0.3045 | 0.0280 | 0.2084 | 128 |
| MeanFlow | 0.0109 | 0.3887 | 0.0347 | 0.3921 | 1 |
| MeanFlow | 0.0091 | 0.3163 | 0.0297 | 0.2422 | 8 |
| MeanFlow | 0.0054 | 0.2722 | 0.0281 | 0.2490 | 32 |
| RMFlow (**ours**) | 0.0059 | 0.2866 | 0.0287 | 0.2499 | 1 |

Table 5: KL divergence between the generated (by different models) and test trajectory distributions, estimated from histogram-based density approximations, with/without conditioning on the event.

### 5.4 TEXT-TO-IMAGE

In this experiment, we train MeanFlow and RMFlow for text-to-image generation on the COCO dataset Chen et al. (2015). Following Stable Diffusion Rombach et al. (2022), all operations are performed in the latent space $\mathbb{R}^{4\times32\times32}$. The mapping $\phi_\omega(c)$ converts the text conditions into initial latent states. Concretely, we fine-tune the pretrained text-embedding model `e5-base` (Wang et al., 2022) and attach an MLP to project the embeddings into the latent space. Additionally, we fine-tune the Stable Diffusion VAE decoder on COCO using PEFT (Hu et al., 2022; Dettmers et al., 2023) so that it can decode the final latent state into images. Both MeanFlow and RMFlow use a 480M-parameter U-Net as the latent-space backbone.

We adopt the Karpathy split (Karpathy & Fei-Fei, 2015) for training and validation, and evaluation is performed with COCO FID-30K following Rombach et al. (2022); He et al. (2025) (details in Appendix B.4). As shown in Table 6, RMFlow attains FID comparable to the best single-step generators on COCO, such as Distilled Stable Diffusion (Liu et al., 2023b), StyleGAN-T (Sauer et al., 2023). Importantly, RMFlow (and MeanFlow) is orthogonal to the other methods listed in Table 6, as it does not rely on auxiliary models for training. In contrast, GAN-based approaches require a discriminator, and distilled models depend on a pretrained teacher. Moreover, our models are trained under limited computational resources (e.g., RTX 3090/4090 GPUs with 24 GB memory) using mixed-precision bf16, whereas most state-of-the-art models listed in Table 6 are trained on multiple A100 80 GB GPUs with full-precision fp16. These results indicate that RMFlow has strong potential for further improvement if trained with larger computational budgets. We also report the performance on CLIP score, see Appendix B.7.

## 6 CONCLUDING REMARKS

In this work, we introduce RMFlow, a refinement of MeanFlow with minimal computational and memory overhead. The central innovation lies in augmenting the 1-NFE MeanFlow with a subsequent noise injection step, which facilitates likelihood maximization. To support this mechanism, we

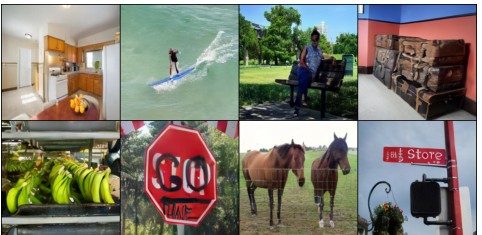 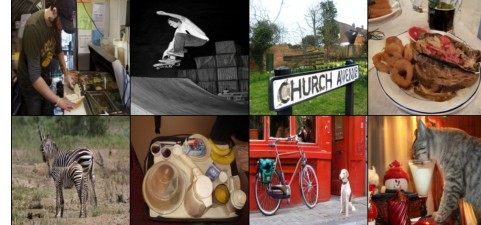

(1) The dining table near the kitchen has a ...
(2) A woman riding a surfboard on a wave in the ...
(3) A woman sitting on a wooden park bench ...
(4) A stack of old trunks and luggage against ...
(5) Rows of unripe bananas on display in ...
(6) stop sign with spray painted words on it.
(7) A couple of horses that are next to a fence.
(8) A red and white street sign mounted on ...

(1) A man making a sandwich on a lunch truck.
(2) A skateboarder performing a trick on an indoor ramp.
(3) There is a large sign that says a street name on it.
(4) A white plate topped with onion rings and ...
(5) A big zebra and a little zebra standing and looking.
(6) The meal is ready on the tray to be eaten.
(7) A bike and a dog on the sidewalk outside a ...
(8) A cat up on a desk drinking milk from a glass.

Figure 5: COCO dataset samples generated using 1-NFE RMFlow conditioned on different input prompts.

| | Type | params | NFE | Teacher-free (or discriminator-free) | COCO FID-30K ($\downarrow$) | Resolution |
|---|---|---|---|---|---|---|
| Stable Diffusion v1.5 Rombach et al. (2022) | Diff | 860M | $\gg 1$ | ✓ | 9.62 | $256 \times 256$ |
| Stable Diffusion v2.1 Rombach et al. (2022) | Diff | 860M | $\gg 1$ | ✓ | 13.45 | $256 \times 256$ |
| FlowTok-XL He et al. (2025) | ODE | 698M | $\gg 1$ | ✓ | 10.1 | $256 \times 256$ |
| Show-o Xie et al. (2024) | Diff | 1.3B | $\gg 1$ | ✓ | 9.24 | $256 \times 256$ |
| PixArt Chen et al. (2023) | ODE | 630M | $\gg 1$ | ✓ | 7.32 | $256 \times 256$ |
| LDM Rombach et al. (2022) | Diff | 1.4B | $\gg 1$ | ✓ | 12.63 | $256 \times 256$ |
| VQGAN+T Jahn et al. (2021) | GAN | 1.1B | 1 | ✗ | 32.76 | $256 \times 256$ |
| LAFITE Zhou et al. (2022) | GAN | 75M | 1 | ✗ | 26.94 | $256 \times 256$ |
| StyleGAN-T Sauer et al. (2023) | GAN | 1B | 1 | ✗ | 13.90 | $256 \times 256$ |
| InstaFlow Liu et al. (2023b) | ODE | 900M | 1 | ✗ | 13.10 | $512 \times 512$ |
| UFOGen Xu et al. (2024b) | Diff | 900M | 1 | ✗ | 12.78 | $512 \times 512$ |
| Stable Diffusion + Distill Liu et al. (2023b) | Diff | 900M | 1 | ✗ | 34.6 | $256 \times 256$ |
| Rectified Flow + Distill Liu et al. (2023b) | ODE | 900M | 1 | ✗ | 20.0 | $256 \times 256$ |
| MeanFlow | ODE | 620M | 1 | ✓ | 27.31 | $256 \times 256$ |
| RMFlow (**ours**) | Diff | 620M | 1 | ✓ | 18.91 | $256 \times 256$ |

Table 6: FID of the generated images on the benchmark COCO2014 dataset using different models.

propose a novel loss function that jointly minimizes the discrepancy between the exact and learned probability paths while maximizing likelihood. Empirical results demonstrate that 1-NFE RMFlow achieves strong performance in multimodal generation tasks.

A promising direction for future research is to extend RMFlow to support multiple mean flow transport steps. Specifically, we envision applying a noise-injection step after each transport step, which would require the design of a corresponding loss function to maintain likelihood maximization. This extension presents additional challenges compared to the current formulation and opens avenues for more expressive and accurate generative modeling. Another limitation of RMFlow is that it uses a fixed parameter $\sqrt{\sigma_{\min}^2 - \sigma^2}$ during the noise injection step, which may be suboptimal. As future work, we plan to explore more adaptive strategies for selecting this parameter, such as making it learnable or following a dynamic schedule.

# 7 ACKNOWLEDGEMENT

This material is based on research sponsored by NSF grants DMS-2152717, DMS-2208361, DMS-2219956, DMS-2436343, and DMS-2436344, and DOE grants DE-SC0023490, DE-SC0025589, and DE-SC0025801.

## ETHICS STATEMENT

In this paper, we propose a new framework to improve MeanFlow for efficient data generation. The new model can generate high-fidelity data efficiently. Our work belongs to fundamental research and is expected to improve existing models for generative modeling. Our work is methodological, and we validate our proposed approaches on the benchmark datasets. We do not expect to cause negative societal problems. Furthermore, we do not see any issues with potential conflicts of interest and sponsorship, discrimination/bias/fairness concerns, privacy and security issues, legal compliance, and research integrity issues (e.g., IRB, documentation, research ethics.

## REPRODUCIBILITY STATEMENT

We are committed to conducting reproducible research. To ensure the integrity and transparency of our work, we employ a multifaceted approach: First, we meticulously compare the novelty of our research against existing literature. This involves a thorough examination of the current state of the field to identify gaps in knowledge and demonstrate the unique contributions of our work. Second, we provide detailed derivations of our proposed approaches and theoretical results. By carefully outlining the mathematical underpinnings of our methods, we enhance the understanding of our work and facilitate its verification by others. Third, we conduct rigorous experiments using widely recognized benchmark datasets. This allows us to evaluate the performance of our methods against established standards and provides a solid foundation for comparison with other approaches. Fourth, we meticulously report experimental details, including the specific datasets used, parameters chosen, and evaluation metrics employed. Finally, we make all experimental codes, accompanied by comprehensive documentation, publicly available. This open-source approach empowers researchers to inspect our methods, verify our results, and build upon our work. By sharing our code, we foster collaboration, advance the field, and contribute to the overall reproducibility of scientific research.

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

## A    TECHNICAL PROOFS

**Theorem 4.1.** *The negative log-likelihood loss $\mathcal{L}_{\text{NLL}}$ provides a lower bound on the expected log-likelihood of the target distribution:*

$$-A \cdot \mathcal{L}_{\text{NLL}} + C \leq \mathbb{E}_{\boldsymbol{x}_{\text{tgt}}}[\log p_\theta(\boldsymbol{x}_{\text{tgt}})] = -H(p_{\text{tgt}}) - D_{\text{KL}}(p_{\text{tgt}} \| p_\theta), \tag{11}$$

*where $H(p_{\text{tgt}}) := -\mathbb{E}_{\boldsymbol{x}_{\text{tgt}}}[\log p_{\text{tgt}}]$ is the entropy of $p_{\text{tgt}}$, $D_{\text{KL}}(p_{\text{tgt}} \| p_\theta) := \mathbb{E}_{\boldsymbol{x}_{\text{tgt}}}[\log \frac{p_{\text{tgt}}}{p_\theta}]$ denotes the KL divergence between the target and the learned distributions, and $A, B > 0$ are constants.*

*proof of Theorem 4.1.* We begin with the marginal likelihood:

$$\begin{aligned}
\log p_\theta(\boldsymbol{x}_{\text{tgt}}) &= \log \mathbb{E}_{\boldsymbol{x}_0}\big[p_\theta(\boldsymbol{x}_{\text{tgt}}|\boldsymbol{x}_0))\big] \\
&\geq \mathbb{E}_{\boldsymbol{x}_0}\big[\log p_\theta(\boldsymbol{x}_{\text{tgt}}|\boldsymbol{x}_0)\big],
\end{aligned} \tag{13}$$

where the inequality follows from Jensen's inequality.

Taking expectation over $\boldsymbol{x}_{\text{tgt}}$ gives

$$\mathbb{E}_{\boldsymbol{x}_{\text{tgt}}}[\log p_\theta(\boldsymbol{x}_{\text{tgt}})] \geq \mathbb{E}_{\boldsymbol{x}_0,\boldsymbol{x}_{\text{tgt}}}\big[\log p_\theta(\boldsymbol{x}_{\text{tgt}}|\boldsymbol{x}_0)\big]. \tag{14}$$

Now, by substituting the log-likelihood expression equation 9, we obtain

$$\begin{aligned}
\mathbb{E}_{\boldsymbol{x}_0,\boldsymbol{x}_{\text{tgt}}}\big[\log p_\theta(\boldsymbol{x}_{\text{tgt}}|\boldsymbol{x}_0)\big] &= \mathbb{E}_{\boldsymbol{x}_0,\boldsymbol{x}_{\text{tgt}}}\Big[-\frac{1}{2(\sigma_{\min}-\sigma)^2}\big\|\boldsymbol{x}_{\text{tgt}} - \big(\boldsymbol{x}_0 + \hat{\mathbf{u}}_{0,1}(\boldsymbol{x}_0;\theta)\big)\big\|^2 + C\Big] \\
&= -\frac{1}{2(\sigma_{\min}-\sigma)^2}\mathbb{E}_{\boldsymbol{x}_0,\boldsymbol{x}_{\text{tgt}}}\Big[\big\|\boldsymbol{x}_{\text{tgt}} - \big(\boldsymbol{x}_0 + \hat{\mathbf{u}}_{0,1}(\boldsymbol{x}_0;\theta)\big)\big\|^2\Big] + C \\
&= -\frac{1}{2(\sigma_{\min}-\sigma)^2}\mathcal{L}_{\text{NLL}} + C
\end{aligned} \tag{15}$$

Combining the inequalities, there exist constants $A, C > 0$ such that

$$-A \cdot \mathcal{L}_{\text{NLL}} + C \ \leq \ \mathbb{E}_{\boldsymbol{x}_{\text{tgt}}}[\log p_\theta(\boldsymbol{x}_{\text{tgt}})]. \tag{16}$$

Finally, recall that

$$\mathbb{E}_{\boldsymbol{x}_{\text{tgt}}}[\log p_\theta(\boldsymbol{x}_{\text{tgt}})] = -H(p_{\text{tgt}}) - D_{\text{KL}}(p_{\text{tgt}} \| p_\theta), \tag{17}$$

which establishes the desired relation.

$\square$

## B    EXPERIMENT SETUP

**Flow map design.**    For a given pair $\boldsymbol{z} = (\boldsymbol{x}_0, \boldsymbol{x}_1)$, we choose the conditional velocity field $\boldsymbol{u}_t(\boldsymbol{x}|\boldsymbol{z})$ following (Albergo & Vanden-Eijnden, 2023), i.e.,

$$\boldsymbol{u}_t(\boldsymbol{x}|\boldsymbol{z}) = \frac{\dot{\gamma}(t)}{\gamma(t)}\big(\boldsymbol{x} - t\boldsymbol{x}_1 - (1-t)\boldsymbol{x}_0\big) + (\boldsymbol{x}_1 - \boldsymbol{x}_0), \tag{18}$$

where $\gamma(t) = \eta(1-t)$ with $\gamma(0) = \eta$, $\gamma(1) = 0$, and $\eta = 10^{-2}, 5 \times 10^{-2}, 10^{-1}$.

**Training Setup**:

*Loss Metric*: Following Geng et al. (2025), we focus on part I (mean flow loss) of $\mathcal{L}_{\text{RMFlow}}$ 12, expressed as $\mathcal{L} = |\Delta|_2^{2\zeta}$, where $\Delta$ denotes the regression error. In practice, we apply a weight $w = \left(\frac{1}{\|\Delta\|_2^2 + 1e-3}\right)^m$ with $m = 1 - \zeta$. When $m = 0.5$, this formulation becomes closely related to the Pseudo-Huber loss introduced in Song & Dhariwal (2023); hence, we adopt $m = 0.5$ for all experiments. The hyperparameters $\lambda_1$ and $\lambda_2$ in $\mathcal{L}_{\text{RMFlow}}$ are selected individually for each experiment, as reported below.

*Time sampling and condition*: For the part III and II in $\mathcal{L}_{\text{RMFlow}}$ 12, we only need to sample $x_0$ so that the time is always zero. Similar one sampling method used in Geng et al. (2025), we sample $(t, r)$ such that $p(t) = 2t$ and

$$p(r|t) = q \cdot \frac{\mathbf{1}_{0 \leq r < t}}{t} + (1 - q)\delta(r - t)$$

so for given sampled $t$, we sample $r$ from $\mathcal{U}[0, t)$ with probability $q$ and set $r = t$ probability $1 - q$. $q$ is selected individually for each experiment, as reported below. We use positional embedding for $(r, t)$, which are then combined and provided as the conditioning of the neural network. As used in Geng et al. (2025), it is not necessary for the network to directly condition on (r, t), so we have $\boldsymbol{u}_{t,r}(\cdot; \theta) := \text{net}(\cdot, t, t - r)$.

## B.1 ABLATION STUDY

In this task, we focus on how the value of $\lambda_1$ balances the Wasserstein (part I in equation 12) and the likelihood (part II in equation 12).

### B.1.1 GAUSSIAN MIXTURE

Table 7 shows that the RMFlow has the best performance when $\lambda_1 = 1e - 1, 1e - 2$.

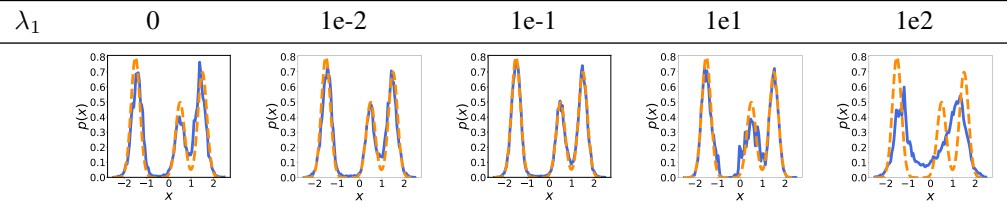

Table 7: Ablation study on the Gaussian mixture task with varying $\lambda_1$.

## B.2 CHECKERBOARD

Table 8 shows that the RMFlow has the best performance when $\lambda_1 = 1e - 1$.

| $\lambda_1$ | 0 | 1e-2 | 1e-1 | 1e1 | 1e2 |
|---|---|---|---|---|---|
| TV | 0.238 | 0.201 | 0.173 | 0.222 | 0.289 |
| KL | 0.311 | 0.228 | 0.163 | 0.237 | 0.425 |

Table 8: Ablation study on the 2D Checkerborad synthetic task with varying $\lambda_1$.

## B.3 QM9

Table 9 shows that the RMFlow has the best performance when $\lambda_1 = 5e - 2$.

| $\lambda_1$ | 0 | 1e-2 | 5e-2 | 1e-1 | 1 | 1e1 |
|---|---|---|---|---|---|---|
| Atomic Stab. | 98.4 | 98.8 | 98.9 | 98.8 | 98.0 | 97.6 |
| Mol Stab. | 84.3 | 92.1 | 93.2 | 92.9 | 87.2 | 83.9 |

Table 9: Ablation study on the QM9 molecule generation task with varying $\lambda_1$.

### B.3.1 CONFIGURATION

The Gaussian mixture and checkerboard experiments share the same configuration, differing only in the model's input and output dimensions. The configuration is summarized in Table 10.

| Model | number of layers | 6 |
|---|---|---|
| | hidden dim | 256 |
| | activation | SiLU |
| Train | iteration | 1e5 |
| | batch size | 256 |
| | optimizer | Adam Diederik (2014) |
| | lr schedule | polynomial |
| | lr | 1e-4 |
| | Adam$(\beta_1, \beta_2)$ | (0.9, 0.95) |
| | ema decay | 0.9995 |
| | precision | fp32 |
| | $\lambda_1$ | [1e-1, 1e-2] |

Table 10: Training Setup and Backbone Configuration for MeanFlow and RMFlow on Synthetic Tasks

### B.4 TEXT-TO-IMAGE

**VAE Decoder**: We use the VAE in `stabilityai/sd-vae-ft-mse` Rombach et al. (2022) and fine-tune the `conv-in`, `up-blocks` (0,-1), `conv-out`, `quant-conv`, and `post-quant-conv` parts of the VAE decoder to refine the decoded image quality on the target dataset using the VAE decoder reconstruction loss Kingma & Welling (2013) with the learning rate $1e-5$.

**Text Embedding Model**: Following Section 5.4, we optimize solely the MLP parameters $\omega$ that map pretrained `e5` embeddings to the latent space under equation 12. Accordingly, $\phi_\omega$ is implemented as a fixed `e5` encoder plus a trainable MLP (i.e., a PEFT setup).

**U-Net Backbone and Training Setup** We build the latent-space backbone for Mean Flow and RM-Flow by reusing selected U-Net blocks ($\sim$480M parameters) from pretrained Stable Diffusion Rombach et al. (2022), with an additional time embedding for $r$. This is effectively transfer learning. Table 11 shows the U-Net configuration and the training setup.

| U-Net | block-out-channels | [320, 640, 1280, 1280] |
|---|---|---|
| | down-block types | [CrossAttnDownBlock2D, CrossAttnDownBlock2D, CrossAttnDownBlock2D, DownBlock2D] |
| | layers-per-block | 2 |
| | attention-head-dim | 8 |
| | cross-attention-dim | 768 |
| **Pre-train** | loss | $\mathcal{L}_{\mathrm{CMFM}}$ |
| | epochs | 500 |
| | batch size per GPU | 16 |
| | optimizer | Adam |
| | lr schedule | polynomial |
| | Warm up epoch | 2 |
| | lr | 1e-4 |
| | Adam$(\beta_1, \beta_2)$ | (0.9, 0.95) |
| | ema decay | 0.9995 |
| | precision | fp16 |
| | trainable param | 480M |
| **Post-train** | loss | $\mathcal{L}_{\mathrm{RMFlow}}$ |
| | epochs | 500 |
| | batch size per GPU | 16 |
| | optimizer | Adam |
| | lr schedule | polynomial |
| | Warm up epoch | 10 |
| | lr | 5e-5 |
| | Adam$(\beta_1, \beta_2)$ | (0.9, 0.95) |
| | ema decay | 0.9995 |
| | precision | bf16 |
| | trainable param | 210M |
| | $\lambda_1$ | [5e-2, 1e-2] |
| Reg for $\phi_\omega$ | $\lambda_2$ | 1e-4 |
| Time sample | $p(r \neq t)$ | 0.25 |

Table 11: Training Setup and Backbone Configuration for MeanFlow and RMFlow on the COCO Text-to-Image Dataset

### B.5 CONTEXT-TO-MOLECULE

**Context Embedding Model**: We implement $\phi_\omega$ as an MLP followed by a single EGNN layer. The MLP projects the context vector into the data space, and the EGNN layer further refines these representations. The EGNN configuration matches that used in the Mean Flow/RMFlow backbone.

**EGNN Backbone and Training Setup** We use the same EGNN Backbone in (Hoogeboom et al., 2022) augmented with a time-embedding module for the additional scalar time variable $r$. Table 12 shows the training setup and configuration.

| EGNN | number of layers | 9 |
|---|---|---|
| | acitivation | SiLU |
| | hidden dim | 256 |
| **Pre-train** | loss | $\mathcal{L}_{\mathrm{CMFM}}$ |
| | epochs | 1500 |
| | batch size per GPU | 64 |
| | optimizer | Adam |
| | lr schedule | polynomial |
| | Warm up epoch | 10 |
| | lr | 1e-4 |
| | Adam$(\beta_1, \beta_2)$ | (0.9, 0.95) |
| | ema decay | 0.9995 |
| | precision | fp32 |
| **Post-train** | loss | $\mathcal{L}_{\mathrm{RMFlow}}$ |
| | epochs | 1500 |
| | batch size per GPU | 64 |
| | optimizer | Adam |
| | lr schedule | polynomial |
| | Warm up epoch | 10 |
| | lr | 1e-4 |
| | Adam$(\beta_1, \beta_2)$ | (0.9, 0.95) |
| | ema decay | 0.9995 |
| | precision | fp16 |
| | $\lambda_1$ | [1e-2, 5e-2] |
| Reg for $\phi_\omega$ | $\lambda_2$ | 1e-4 |
| Time sample | $p(r \neq t)$ | 0.5 |

Table 12: Training Setup and Backbone Configuration for MeanFlow and RMFlow on Context-to-Molecule Generation on QM9 Dataset

### B.6 TIME SERIES: DYNAMIC SYSTEM

**Trajectory dataset**: we use the same dataset as decribed in (Huang et al., 2025, Appendix B.1).

**Models**: (1) We implement the guidance embedding function $\phi_\omega$ as an MLP that maps the event-guidance vector together with the first three states $\boldsymbol{x}(\tau_1), \boldsymbol{x}(\tau_2), \boldsymbol{x}(\tau_3)$ into the prior sample; (2) we adopt the UNet architecture from Finzi et al. (2023), adding an additional time embedding for $r$.

**Training Setup**: see Table 13

| Train | iteration | 1e5 |
|---|---|---|
| | batch size | 500 |
| | optimizer | Adam |
| | lr | 1e-4 |
| | weight decay | 0.995 |
| | $\lambda_1$ | 1e-1 |
| | $\lambda_2$ | 1e-4 |

Table 13: Training Setup for MeanFlow and RMFlow on Dynamical System Forecasting Tasks

### B.7 ADDITIONAL EXPERIMENTS

In this section, we report the comparison of CLIP score of our RMFlow, MeanFlow, and some models on the COCO dataset (2017 version dataset splitting, with 5000 images, following Liu et al. (2023b)). See Table 14.

| | Type | params | NFE | Teacher-free (or discriminator-free) | FID-5k | Clip (↑) |
|---|---|---|---|---|---|---|
| Stable Diffusion v1.5 Rombach et al. (2022) | Diff | 860M | ≫ 1 | ✓ | 20.1 | 0.315 |
| InstaFlow Liu et al. (2023b) | ODE | 900M | 1 | ✗ | 23.4 | 0.304 |
| StyleGAN-T Sauer et al. (2023) | GAN | 1B | 1 | ✗ | 24.1 | 0.305 |
| PD-SD Meng et al. (2023) | Diff | N/A | 1 | ✗ | 37.2 | 0.275 |
| MeanFlow | ODE | 620M | 1 | ✓ | 38.5 | 0.273 |
| RMFlow (**ours**) | Diff | 620M | 1 | ✓ | 27.9 | 0.291 |

Table 14: CLIP scores of the generated images on the benchmark COCO2017 dataset using different models.

## B.8 POLICY GRADIENT WITH PHYSICAL FEEDBACK

We define a reward $r(\hat{\boldsymbol{x}}_{\text{tgt}})$ to quantify the molecule stability of the generated sample $\hat{\boldsymbol{x}}_{\text{tgt}}$ on QM9 dataset, then define the policy gradient following Black et al. (2023):

$$\nabla_\theta \mathcal{L}_{\text{RL}} := \mathbb{E}\Big[\nabla_\theta \log p_\theta(\boldsymbol{x}_{\text{tgt}} \mid \boldsymbol{x}_0) r(\hat{\boldsymbol{x}}_{\text{tgt}})\Big]$$

and the corresponding loss function

$$\mathcal{L}_{\text{RL}} := -\mathbb{E}\Big[\log p_\theta(\boldsymbol{x}_{\text{tgt}} \mid \boldsymbol{x}_0) r(\hat{\boldsymbol{x}}_{\text{tgt}})\Big]$$

where

$$\log p_\theta(\boldsymbol{x}_{\text{tgt}} \mid \boldsymbol{x}_0) = -\frac{1}{2(\sigma_{\min}^2 - \sigma^2)}\big\|\boldsymbol{x}_{\text{tgt}} - \big(\boldsymbol{x}_0 + \hat{\boldsymbol{u}}_{0,1}(\boldsymbol{x}_0;\theta)\big)\big\|^2 + C,$$

We then perform reinforcement learning fine-tuning by augmenting the RMFlow objective 12:

$$\mathcal{L}_{\text{RMF+RL}}(\theta, \omega) = \mathcal{L}_{\text{RMFlow}}(\theta, \omega) + \eta \mathcal{L}_{\text{RL}}(\theta)$$

where $\eta$ is a hyperparameter.

