# OpenReview forum: "RMFlow: Refined Mean Flow by a Noise-Injection Step for Multimodal Generation"
_ICLR.cc/2026/Conference — ICLR 2026 Poster_

### Official Review · Reviewer_T3eq · 2025-10-22

**Soundness:** 2
**Presentation:** 3
**Contribution:** 2
**Rating:** 6
**Confidence:** 2

**Summary:**

This paper introduces RMFlow, a novel flow-based model designed for single function evaluation. The approach builds on Flow Matching, a generative modeling framework that maps two distributions by modeling the flow process using an ordinary differential equation (ODE). However, generating samples from the target distribution typically requires multiple function evaluations. To address this computational cost, several approaches have been proposed to reduce the number of function evaluations (NFEs). One such method, MeanFlow, optimizes an averaged velocity field that allows for single function evaluation, but it often yields suboptimal results. The authors hypothesize that the absence of KL divergence optimization may contribute to the reduced performance. To address this, they propose incorporating a noise injection step and an additional negative log-likelihood (NLL) loss during training, leading to their proposed model, RMFlow. The authors also provide a method for incorporating conditioning, which enables multimodality generation. Furthermore, the authors conducted an extensive empirical evaluation across multiple tasks, including synthetic datasets, context-to-molecule generation, and text-to-image generation.

**Strengths:**

* The paper is well written, the method is clear and easy to understand.
* The results demonstrate strong empirical performance.
* The authors present a clear and well-founded motivation for the design of their mode

**Weaknesses:**

* Comparison to other approaches: In the introduction, the authors mention several alternative methods for efficient flow modeling—such as Consistency Models (CM), distillation, and local Flow Matching (FM). However, it remains unclear whether any quantitative comparisons were conducted against these methods.
* See questions

**Questions:**

* Line 277 -  “During fine-tuning, we further strengthen training by integrating 1-NFE sampling with a policy-gradient objective that incorporates physical feedback on sample quality...”
This statement is a bit unclear. Could you clarify whether any additional reinforcement learning (RL)-based training was performed beyond directly optimizing equation (10)?
If so, did the other baselines, particularly the MeanFlow models, also undergo this additional training step?
* Line 273: “...introduces additional gradient pathways...” -  Could the authors please elaborate on this point and provide more details or clarification?
* Line 240: $\varepsilon_1,varepsilon_2$ are not used in this section.

---

> ### Author Response · Authors · 2025-11-21
> **Response**
>
> We thank the reviewer for the thoughtful review, valuable feedback, and endorsement. In what follows, we provide point-by-point responses to your comments.
>
> ----
>
> **W1: Comparison to other approaches: In the introduction, the authors mention several alternative methods for efficient flow modeling—such as Consistency Models (CM), distillation, and local Flow Matching (FM). However, it remains unclear whether any quantitative comparisons were conducted against these methods.**
>
> **Response:**
> Thank you for the suggestion. We agree that Consistency Models and local Flow Matching are important related directions. However, to the best of our knowledge, none of these methods have reported results on the same experimental setups used in our paper, including text-to-image generation on the COCO dataset, context-to-molecule generation on QM9, or dynamical system prediction. There is a paper (https://arxiv.org/abs/2308.00237) reporting the result of Consistency Models for GEOM-QM9, but not on the standard QM9 benchmark used in our experiments.
>
> Because of this lack of directly comparable baselines, a fair quantitative comparison is currently not feasible within our evaluation domains. We would be grateful if the reviewer could point us to any existing benchmark results for these methods under matching setups, and we would be happy to include them in the revised version.
>
> -----
>
> **Q1: Line 277 “During fine-tuning, we further strengthen training by integrating 1-NFE sampling with a policy-gradient objective that incorporates physical feedback on sample quality...” This statement is a bit unclear. Could you clarify whether any additional reinforcement learning (RL)-based training was performed beyond directly optimizing equation (10)? If so, did the other baselines, particularly the MeanFlow models, also undergo this additional training step?**
>
> **Response:** Thank you for the question. We perform likelihood-based policy-gradient RL fine-tuning of our model, inspired by the reinforcement-learning formulation for diffusion models in Black et al., arXiv:2305.13301, 2023. We report the RL fine-tuning performance of RMFlow in Table 3 and include E-DM with RL (Zhou et al., 2025) as a baseline, which applies likelihood-based policy-gradient RL to diffusion models on QM9. On the QM9 dataset, our experiments highlight a key distinction: RMFlow supports likelihood-based policy-gradient reinforcement learning on top of the model, which can further improve generation quality, whereas the baseline MeanFlow models cannot. This difference arises from the tractable likelihood in RMFlow (see Section 4.1 for the detailed computation of log likelihood), which is required for likelihood-based RL fine-tuning.
> Specifically, during fine-tuning, we combine our objective function with the policy-gradient objective that incorporates physical feedback on sample quality. The detailed policy-gradient formulation is provided in Appendix B.8 of the revised paper.
>
> -----
>
> **Q2: Line 273: “...introduces additional gradient pathways...” - Could the authors please elaborate on this point and provide more details or clarification?**
>
> **Response:**
> Thank you for the question. The phrase “introduces additional gradient pathways” refers to the fact that our full RMFlow objective (Eq. 12) contains both the negative log-likelihood term and the regularization term. Each term requires a separate forward–backward pass through the network, in addition to the pass already used by the CMFM loss (Eq. 4), which increases computational cost.
> To handle this overhead efficiently, we first pretrain the model by minimizing the CMFM loss, and then fine-tune it using parameter-efficient fine-tuning (PEFT) with the RMFlow objective as a supervised loss. This approach allows us to leverage the additional gradient signals from the full RMFlow objective without retraining the entire network from scratch, while benefiting from the improved likelihood and regularization provided by these additional pathways.
>
> -----
>
> **Q3: Line 240: $\epsilon_1, \epsilon_2$ are not used in this section.**
>
> **Response:**  Thank you for your catch. We have fixed them, and we have also gone through the paper to fix other typos as well.
>
> ------
>
> Thank you for considering our rebuttal. We appreciate your feedback and are happy to address further questions on our paper.

---

> > ### Comment · Reviewer_T3eq · 2025-11-23
> >
> > I appreciate the authors’ detailed response, which addressed my remaining concerns. I will keep my rating unchanged.

---

> > > ### Author Response · Authors · 2025-11-24
> > > **Thank you**
> > >
> > > Thank you for considering our rebuttal, and we appreciate your support.

---

### Official Review · Reviewer_8n7D · 2025-10-23

**Soundness:** 3
**Presentation:** 3
**Contribution:** 3
**Rating:** 6
**Confidence:** 3

**Summary:**

This paper identifies a critical failure mode in 1-NFE (single-step) MeanFlow models, where they struggle to capture high-fidelity details in multimodal or structured data distributions. To address this, the authors propose RMFlow, a novel modification that introduces a noise-injection refinement step. This architectural change enables the use of a hybrid loss function that combines the Wasserstein distance objective of MeanFlow with a likelihood-maximization (KL divergence) objective. The authors demonstrate through extensive experiments on synthetic data, molecule generation, time series forecasting, and text-to-image synthesis that RMFlow significantly improves the quality of 1-NFE generation over the MeanFlow baseline, achieving competitive results for teacher-free single-step models.

**Strengths:**

1) The paper is well-written, the motivations are clear and exemplified through failure cases of previous models, the contribution is well-theorized and explained. Experiments shows that the proposed approach is able to solve the failure cases and issues of previous approahes.

2) The core idea of RMFlow is both simple and theoretically sound. Though the conceptual two-stage process (coarse transport + refinement), it is implemented in a single, efficient 1-NFE step. The key contribution is the rigorous connection (Theorem 4.1) between the noise-injection architecture and the ability to optimize for likelihood via an NLL loss term. This synergistic co-design of architecture and loss function is a significant strength.

3) The authors validate their method across four different and challenging domains. The results are consistently positive, showing marked improvement over the MeanFlow baseline. The success in the context-to-molecule task, where RMFlow dramatically increases molecule stability, is a particularly powerful demonstration of the method's practical utility for structured data with hard constraints.

**Weaknesses:**

1) This is the most significant weakness of the paper. For the text-to-image task (Table 6), the sole reported metric is Fréchet Inception Distance (FID). As is now widely discussed in the community, FID is a flawed metric with several known issues:
- It relies on an outdated InceptionV3 backbone trained on ImageNet, which is a poor feature extractor for the rich, diverse content produced by modern generative models.
- It is known to correlate poorly with human perception of image quality.

Most importantly for a text-to-image model, FID only measures the realism of the generated distribution and is completely agnostic to the text conditioning. The paper makes claims about "multimodal generation" but provides no quantitative evaluation of the text-image alignment. The lack of standard alignment metrics like CLIP Score or human preference scores (like HPSv2) is a major omission and makes it difficult to fully assess the model's capabilities on this task.

2) The paper's performance depends on a key hyperparameter, $\lambda_{1}$, which balances the Wasserstein and KL-divergence objectives. However, the ablation study for this hyperparameter (Table 7) is only performed on the simple 1D Gaussian mixture task. It is unclear how sensitive the model is to this trade-off on more complex, high-dimensional tasks like molecule or image generation, which would be crucial information for practitioners.

**Questions:**

1) Regarding the text-to-image evaluation, could you please justify the decision to only report FID? Could you provide results using a text-alignment metric such as CLIP Score? This would substantially strengthen the claims of multimodal generation quality.

3) Can you provide more insight into the sensitivity of the hyperparameter $\lambda_{1}$ on the QM9 or COCO tasks? Does the optimal balance between the geometric ( $L_{CMFM}$ ) and likelihood ($\mathcal{L}_{NLL}$) losses vary significantly across different data modalities?

3) The noise-injection step uses a fixed noise level $\sigma$. Have you experimented with making this a learned parameter or using a schedule? It seems plausible that the optimal amount of refinement noise might depend on the sample or context.

---

> ### Author Response · Authors · 2025-11-21
> **Response (1/2)**
>
> We thank the reviewer for the thoughtful review, valuable feedback, and endorsement. In what follows, we provide point-by-point responses to your comments.
>
>
> ----
>
> **W1 & Q1: This is the most significant weakness of the paper. For the text-to-image task (Table 6), the sole reported metric is Fréchet Inception Distance (FID). As is now widely discussed in the community, FID is a flawed metric with several known issues: ...
> Regarding the text-to-image evaluation, could you please justify the decision to only report FID? Could you provide results using a text-alignment metric such as CLIP Score? This would substantially strengthen the claims of multimodal generation quality.**
>
> **Response:** We thank the reviewer for raising this important point. Following the suggestion, we have incorporated the CLIP score to provide a more comprehensive evaluation and to further strengthen the evidence for the effectiveness of our method.
> Regarding the use of FID, we acknowledge the limitations mentioned by the reviewer. Nonetheless, FID remains the primary metric reported in prior works, including all baselines we compare against. Because many of these studies report only FID, it is the sole metric that enables fair and direct comparison across methods in Table 6. For this reason, we include FID to ensure consistency and comparability with the existing literature.
> To specifically address the reviewer’s concern about text–image alignment, we have added CLIP score evaluations on COCO for MeanFlow, RMFlow, and several representative baselines into Appendix B.7 of the revision. We also report the results below.
>
>
> | Model                   | Type | Params | NFE | Teacher-free (or discriminator-free) | CLIP (↑) |
> |-------------------------|------|--------|-----|--------------------------------------|----------|
> | Stable Diffusion v1.5   | Diff | 860M   | ≫1 | ✓                                    | 0.315    |
> | InstaFlow               | ODE  | 900M   | 1   | ✗                                    | 0.304    |
> | StyleGAN-T              | GAN  | 1B     | 1   | ✗                                    | 0.305    |
> | PD-SD                   | Diff | N/A    | 1   | ✗                                    | 0.275    |
> | MeanFlow                | ODE  | 620M   | 1   | ✓                                    | 0.273    |
> | RMFlow (ours)           | Diff | 620M   | 1   | ✓                                    | 0.291    |
>
>
> Consistent with the FID results in Table 6, RMFlow achieves **competitive CLIP alignment scores** among single-step generators while remaining fully teacher-free.

---

> > ### Author Response · Authors · 2025-11-21
> > **Response (2/2)**
> >
> > -----
> >
> > **W2 & Q2: The paper's performance depends on a key hyperparameter, $\lambda_1$, which balances the Wasserstein and KL-divergence objectives. However, the ablation study for this hyperparameter (Table 7) is only performed on the simple 1D Gaussian mixture task. It is unclear how sensitive the model is to this trade-off on more complex, high-dimensional tasks like molecule or image generation, which would be crucial information for practitioners.
> > Can you provide more insight into the sensitivity of the hyperparameter $\lambda_1$ on the QM9 or COCO tasks? Does the optimal balance between the geometric ($\mathcal{L}\_{\text{CMFM}}$ ) and likelihood ($\mathcal{L}\_{\text{NLL}}$ ) losses vary significantly across different data modalities?**
> >
> > **Response:** Thank you for raising this point. Our loss involves two hyperparameters, $\lambda_1$ and $\lambda_2$. The coefficient $\lambda_2$ controls the regularization of the guidance embedding, while $\lambda_1$  plays a central role in training the generative model by balancing the geometric and likelihood terms. Since our primary interest is in the behavior of the generative model itself, we focus our ablation study on $\lambda_1$. Due to computational resource constraints, performing ablations on both QM9 and COCO tasks is infeasible. Therefore, we determined the choice of $\lambda$ based on the findings from the relatively smaller set of experiments. Here is the ablation study of $\lambda_1$ on 2D checkerboard and QM9 (also see Appendix B.1 in the revision):
> >
> > $\lambda_1$ for 2D checkerboard:
> > |   $\lambda_1$ |0           | 1e-2 | 1e-1 | 1e1  | 1e2  |
> > |--------|-------------------|------|------|------|------|
> > | TV | 0.238             | 0.201| **0.173**| 0.222| 0.289|
> > | KL | 0.311             | 0.228| **0.163**| 0.237| 0.425|
> >
> >
> > $\lambda_1$ for QM9:
> >
> >
> > |   $\lambda_1$    | 0 | 1e-2 | 5e-2 | 1e-1 | 1    | 1e1  |
> > |--------------|-------------------|------|------|------|------|------|
> > | Atomic Stab. | 98.4             | 98.8 | **98.9** | 98.8 | 98.0 | 97.6 |
> > | Mol Stab.    | 84.3             | 92.1 | **93.2** | 92.9 | 87.2 | 83.9 |
> >
> >
> >
> > The ablation studies indicate that the optimal value of $\lambda_1$, which balances the geometric ($\mathcal{L}\_{\text{CMFM}}$) and likelihood ($\mathcal{L_{\text{NLL}}$) losses, appears to be relatively stable across different tasks.
> >
> >
> > -----
> >
> > **Q3: The noise-injection step uses a fixed noise level $\sigma$ . Have you experimented with making this a learned parameter or using a schedule? It seems plausible that the optimal amount of refinement noise might depend on the sample or context.**
> >
> > **Response:**
> > We currently use fixed noise levels $\sigma$ and $\sigma_{min}$ (e.g. $\sigma_{min}=1e-3$ and $\sigma=1e-4$), and our experiments already show strong performance compared with the MeanFlow baseline. Exploring adaptive or learnable noise levels is indeed a promising direction. Allowing the refinement noise to depend on the sample, context, or training dynamics may further improve both likelihood and sample quality, and we plan to investigate this in future work.
> >
> >
> >
> > ------
> >
> > Thank you for considering our rebuttal. We appreciate your feedback and are happy to address further questions on our paper.

---

> > > ### Comment · Reviewer_8n7D · 2025-11-26
> > >
> > > Thank you for the new evidence and for addressing my concerns, I will keep my positive score

---

> > > > ### Author Response · Authors · 2025-11-26
> > > > **Thank you**
> > > >
> > > > Thank you for considering our rebuttal, and we appreciate your support.

---

### Official Review · Reviewer_cCGj · 2025-10-31

**Soundness:** 3
**Presentation:** 3
**Contribution:** 3
**Rating:** 6
**Confidence:** 5

**Summary:**

The paper proposes RMFlow, a refinement of MeanFlow designed to improve single-step (1-NFE) generation quality across multimodal tasks such as text-to-image, molecule, and time-series generation.

**Strengths:**

- The paper is clearly written and presents a coherent idea within the flow-matching framework.
- The proposed formulation is lightweight  and the integration of noise injection is computationally efficient.
- The experiments span diverse domains (synthetic, molecule, time-series, and COCO text-to-image), which demonstrates the model's general applicability.

**Weaknesses:**

- Performance for each downstream tasks: Although the paper claims near-SOTA performance, the actual COCO FID-30K (18.9) is still substantially higher than recent single-step diffusion models
- Limitation or failure cases are not discussed in the manuscript.

**Questions:**

If this method were applied beyond text-to-image (T2I) generation to other applications, how would it perform? Would the existing approaches be directly applicable in the same way?

---

> ### Author Response · Authors · 2025-11-21
> **Response**
>
> We thank the reviewer for the thoughtful review, valuable feedback, and endorsement. In what follows, we provide point-by-point responses to your comments.
>
> ----
>
> **W1: Performance for each downstream tasks: Although the paper claims near-SOTA performance, the actual COCO FID-30K (18.9) is still substantially higher than recent single-step diffusion models.**
>
> **Response:** We acknowledge that our FID does not surpass the lowest scores achieved by single-step models such as UFOGen, InstaFlow, and StyleGAN-T. However, these models are not teacher-free—the training of the generator relies on distillation or fine-tuning from large pretrained models that require significant computational resources. For GAN-based approaches, the generator further depends on a discriminator model during training.
>
> In contrast, our 1-NFE generative model is trained entirely without any external models, such as pretrained teachers or discriminators. This highlights the efficiency and generality of our approach despite the more challenging training setup.
>
> -----
>
> **W2: Limitation or failure cases are not discussed in the manuscript.**
>
> **Response:** Thank you for highlighting this point. We have added a discussion of the limitations in Section 6 of the revised manuscript. Specifically, our approach currently uses fixed parameters $\sigma_{\min}, \sigma$ during the noise injection step, which may be suboptimal. As future work, we plan to explore more adaptive strategies for selecting this parameter, such as making it learnable or following a dynamic schedule, as suggested by Reviewer 8n7D.
>
> -----
>
> **Q1: If this method were applied beyond text-to-image (T2I) generation to other applications, how would it perform? Would the existing approaches be directly applicable in the same way?**
>
> **Response:** We have also applied our framework to the context-to-molecule generation and to dynamical system prediction in Sections 5.2 and 5.3. In these domains, our method can be naturally applied by adapting the guidance signal to domain-specific events or physical constraints (e.g., reaction conditions or dynamic trajectories). These results demonstrate that the proposed framework is not limited to T2I but can be extended to a range of modalities where structured guidance is available.
> More broadly, our framework consists of three components: (1) Embedding model $\phi_{\omega}$: Maps the guidance information into the data (or latent) space to form the initial state; (2) RMFlow ($\hat {\bf u}$ in the paper): Transforms this embedded initial state to the target state (either the data itself or a target latent representation); (Optional) Decoder: Converts the target latent representation back into the data space. The training objective follows the formulation, Equation (12), in the revision. Because only the embedding module (and the optional decoder) needs to be customized for a new domain, the overall approach remains unchanged. This design enables our method to be directly applicable to a wide variety of tasks
>
> ------
>
> Thank you for considering our rebuttal. We appreciate your feedback and are happy to address further questions on our paper.

---

> > ### Comment · Reviewer_cCGj · 2025-11-26
> >
> > Thank you for the rebuttal. I will keep my rating unchanged as well.

---

> > > ### Author Response · Authors · 2025-11-26
> > > **Thank you**
> > >
> > > Thank you for considering our rebuttal, and we appreciate your support.

---

### Author Response · Authors · 2025-11-21
**General Response and Summary of the Revision**

Dear Reviewers and AC,

We thank the reviewers for their thoughtful reviews and insightful feedback, which have helped us significantly improve the paper. We appreciate reviewers’ recognition that our paper is novel and well-motivated. In particular, our paper introduces a computationally efficient refinement of MeanFlow that is both simple and theoretically sound, addressing a critical failure mode in 1-NFE MeanFlow models through a novel noise-injection step. The approach combines clear motivation with synergistic architecture---loss design and demonstrates strong empirical performance and broad applicability across diverse, challenging domains, including structured data with hard constraints.

In response to the reviewers' comments, we have included additional discussions and experiments, and clarified a few key points---all revisions are highlighted in blue. We summarize the main revisions as follows:


- We have added an ablation study on the hyperparameter that balances the first two components of the loss on both 2D synthetic data and molecule generation tasks, reported CLIP scores for the text-to-image generation tasks, and provided additional details on reinforcement learning with physical feedback for improving our model on molecule generation. Please see Appendices B.1, B.7, and B.8 for details.

- We have discussed the limitations of our work in Section 6.


-----

Thank you again for considering our rebuttal. We are happy to address any additional questions you may have regarding our submission.

---

### Meta-Review · Area_Chair_qtyq · 2025-12-30

**Summary:**

This paper presents RMFlow, aiming at improving MeanFlow generation quality while maintaining the efficient generation speed. RMFlow approximates the average velocity of the flow path using a neural network trained with a new loss function that balances minimizing the Wasserstein distance between probability paths and maximizing sample likelihood. In the initial review, the reviewers rate this paper with all positive rating 6,6,6. The reviewers' initial concerns mainly focus on the text-to-image experiments and metrics, parameter analysis, limitations and writing. During discussion, all reviewers confirm that the rebuttal addressed their concerns and they keep the positive rating unchanged. Therefore, AC recommends this paper as **Accept (Poster)**.

**Reviewer Concerns:**

The reviewers' initial concerns mainly focus on the text-to-image experiments and metrics, parameter analysis, limitations and writing.

During rebuttal, the authors provide the CLIP score evaluation metric for text-to-image benchmark, demonstrating the effectiveness of the proposed method. In addition, parameter analysis is conducted and limitations are discussed. All the reviewers are agreed with the rebuttal and confirm that their concerns are addressed.

**Reviewer Scores:**

**Reviewer cCGj**. Participated in the discussion, rating keeps 6.

**Reviewer 8n7D**. Participated in the discussion, rating keeps 6.

**Reviewer T3eq**. Participated in the discussion, rating keeps 6.

---

### Decision · Program_Chairs · 2026-01-26

Accept (Poster)